# SADS-CoV nsp5 Inhibits Interferon Production by Targeting Kinase IKKε

**DOI:** 10.3390/microorganisms13071494

**Published:** 2025-06-26

**Authors:** Gaoli She, Chunhui Zhong, Yue Pan, Zexin Chen, Jingmin Li, Mingchong Li, Yufang Liu, Yongchang Cao, Xiaona Wei, Chunyi Xue

**Affiliations:** 1State Key Laboratory of Biocontrol, School of Life Sciences, Sun Yat-sen University, Guangzhou 510275, China; 2School of Life Sciences, Zhengzhou University, Zhengzhou 450001, China

**Keywords:** SADS-CoV, GDS04, IFN-β, nsp5, IKKε

## Abstract

Swine acute diarrhea syndrome coronavirus (SADS-CoV), initially identified in China in February 2017, severely impacts the swine industry by causing lethal watery diarrhea in neonatal piglets. Understanding the molecular mechanism employed by SADS-CoV to evade the host’s immune defenses is of utmost importance. In this study, using the porcine ileum epithelial cell line IPI-FX as an in vitro model, we investigated the highly pathogenic SADS-CoV GDS04 strain and its nonstructural protein 5 (nsp5) for their roles in inhibiting interferon-beta (IFN-β) production. Our findings indicated that GDS04 inhibited poly(I:C)-induced IFN-β production by impeding the promoter activities of IRF3 and NF-κB. As a 3C-like protease, SADS-CoV nsp5 functioned as an interferon inhibitor by interacting with IKKε, reducing its protein abundance, and inhibiting its phosphorylation. This study enhances our understanding of the interaction between coronaviruses and their hosts, providing novel insights into the evasion of the immune system by coronaviruses.

## 1. Introduction

The *Coronaviridae* family, which consists of enveloped, single-stranded, positive-sense RNA viruses, has gained significant attention following the global outbreaks of severe acute respiratory syndrome coronavirus 2 (SARS-CoV-2). Swine acute diarrhea syndrome coronavirus (SADS-CoV), a bat-HKU2-like coronavirus, was initially identified and reported in Guangdong, China in 2017 [1,2,3] and recently led to an outbreak in Henan, central China in June 2023 [4]. This virus can induce severe watery diarrhea in newborn piglets, resulting in high mortality rates. The genome of SADS-CoV is approximately 27 kb in length and contains 5′UTR-ORF1a/1b-S-NS3a-E-M-N-NS7a-NS7b-3′UTR, encoding four structural proteins (spike (S), envelope (E), membrane (M), and nucleocapsid (N)), 16 non-structural proteins (Nsp1–16), and three accessory proteins (NS3a, NS7a, and NS7b) [2,5]. The interaction between viruses and hosts represents a dynamic arms race, where multiple structural and non-structural viral proteins play pivotal roles in suppressing host innate immunity to facilitate viral replication.

Interferon-beta (IFN-β), as the first line of defense against viruses, plays a critical role in initiating the host’s antiviral responses [6]. During virus infection, the host immune system is activated by pattern recognition receptors (PRRs) that recognize viral pathogen-associated molecular patterns (PAMPs), such as double-stranded RNA (dsRNA). Two major PRRs, retinoic acid-inducible gene I (RIG-I) and melanoma differentiation-associated gene 5 (MDA5), detect dsRNA in the cytoplasm [7,8]. Upon binding with dsRNA, RIG-I and MDA5 recruit the adapter protein IFN-β promoter stimulator 1 (IPS-1), also known as mitochondrial antiviral signaling protein (MAVS). This leads to the activation of TANK-binding kinase 1 (TBK1) and inhibitor of κB kinase-ε (IKKε), followed by the activation and nuclear translocation of the transcription factors IFN regulatory factor 3 (IRF3) and nuclear factor-kappa B (NF-κB). These transcription factors bind to the IFN-β promoter and initiate IFN-β transcription. Consequently, IFN-β secretion induces the expression of IFN-stimulated genes (ISGs), which exert antiviral effector functions [9,10,11]. To counteract the host IFN system, coronaviruses have developed various mechanisms to suppress IFN production, encoding IFN inhibitors to evade the host’s antiviral immunity. Proteins encoded by other coronaviruses, such as SARS-CoV and SARS-CoV-2, have been identified to inhibit IFN production through distinct mechanisms [12,13]. Similarly, 11 proteins encoded by porcine epidemic diarrhea virus (PEDV) have been identified as IFN inhibitors: N, M, E, ORF3, nsp1, nsp3, nsp5, nsp7, nsp14, nsp15, and nsp16 [14].

Coronavirus nsp5, also known as 3C-like protease or main protease, is responsible for processing the viral polyprotein to generate most of the non-structural proteins during viral replication [15,16,17]. Several other small ribonucleic acid viruses, such as foot-and-mouth disease virus (FMDV) [18], hepatitis A virus (HAV) [19], and Seneca Valley virus (SVV) [20,21], have been reported with 3C proteases that target crucial signal molecules in the RIG-I signal pathway, including MDA5, MAVS, RIG-I, IRF7, and IRF9, to inhibit IFN production [16]. Similarly, the 3C-like proteases of PEDV, PDCoV, SARS-CoV, SARS-CoV-2, and SADS-CoV have been reported to impair the innate immune signaling [22,23,24,25]. SARS-CoV-2 nsp5 upregulates SUMOylation of MAVS to activate the NF-κB pathway [26]; MERS-CoV nap5 attenuates IFN-I production by inhibiting IRF3 nuclear translocation [27]; PDCoV nsp5 inhibits IFN-I signaling by cleaving IFIT3, NEMO, and STAT2 [24,28,29]. It can be seen that coronavirus nsp5 can inhibit host IFN production and signal transduction through multiple mechanisms.

Notably, recent studies have shown that SADS-CoV antagonizes IFN-β production by blocking IPS-1 and RIG-I [5]; the N protein inhibits IFN-β production by targeting TBK1 to disturb the interaction between TRAF3 and TBK1 [30]; nsp1 inhibits IFN-β production by preventing TBK1 phosphorylation and inducing CBP degradation [31]; and nsp1 suppresses IFN-λ1 production by degrading IRF1 via the ubiquitin-proteasome pathway [32]. Based on these findings, we propose to investigate whether SADS-CoV nsp5 possesses the function of inhibiting IFN. Our findings revealed that SADS-CoV GDS04 inhibited IFN-β by impeding the activation of IRF3 and NF-κB promoters. Additionally, we confirmed that nsp5 acted as an IFN-β inhibitor by directly interacting with IKKε, leading to a reduction in IKKε protein abundance and inhibition of its phosphorylation. These findings provide new insights into the evasion of host innate immunity by SADS-CoV.

## 2. Materials and Methods

### 2.1. Viruses, Cells, and Reagents

SADS-CoV strain GDS04 (GenBank accession no. MF167434.1) was previously described [33,34]. Viral propagation of GDS04 is trypsin-dependent, with a concentration of 10 μg/mL used in this study. The porcine ileum epithelial cell line, IPI-FX, kindly provided by Prof. Shaobo Xiao (Huazhong Agricultural University, Wuhan, China), was cultured in Dulbecco’s modified Eagle’s medium (DMEM, Gibco, Grand Island, NY, USA) supplemented with 10% fetal bovine serum (FBS, Gibco, Grand Island, NY, USA) at 37 °C with 5% CO_2_. IPI-FX cells were cultured as confluent monolayers and utilized in subsequent experiments. Poly(I:C) (Invitrogen, Carlsbad, CA, USA) was used as inducer of IFN-β and dissolved in water to obtain a stock solution of 1 mg/mL. Anti-IRF3 antibody was purchased from Proteintech (Wuhan, China), anti-IKKε antibody was purchased from Cell Signaling Technology (Danvers, MA, USA), anti-phosphorylated IRF3 (p-IRF3) rabbit monoclonal antibody was purchased from Absin Bioscience (Shanghai, China), anti-GFP and anti-Myc antibodies were obtained from Santa Cruz Biotechnology (Santa Cruz, CA, USA), and anti-HA antibody was obtained from Abmart (Shanghai, China). Anti-SADS-CoV N antibody was prepared and validated by our team.

### 2.2. Plasmids

SADS-CoV nsp5 was cloned into pEGFP-N1 and pcDNA3.1-myc vectors. Porcine IRF3 was constructed into pcDNA3.1-HA vector. Luciferase reporter plasmids (IFN-β-Luc, IRF3-Luc, and NF-κB-Luc) were constructed in the pGL3-Basic vector, with pRL-TK used as an internal control. Porcine IKKε and TBK1 expression plasmids were kindly provided by Prof. Jingyun Ma (South China Agricultural University, Guangzhou, China).

### 2.3. RNA Extraction and Quantitative Real-Time RT-qPCR

Total RNA was extracted using the EZ-press RNA Purification Kit according to the manufacturer’s protocol (EZBioscience, Roseville, CA, USA). The cDNA was synthesized using the HiScript^®^ III RT SuperMix for qPCR (Vazyme, Nanjing, China), and used in an SYBR green PCR assay (Transgen, Beijing, China) with specific primers targeting IFN-β or SADS-CoV N, normalized with GAPDH (Table 1).

### 2.4. Luciferase Reporter Gene Assay

Monolayer IPI-FX cells were cultured in 24-well plates and co-transfected with a specific reporter plasmid (IFN-β-Luc, IRF3-Luc, or NF-κB-Luc), pRL-TK (utilized as an internal control), and the GDS04 nsp5 expression plasmid once the cells reached approximately 80% confluence. After 24 h, the cells were transfected with poly(I:C). Subsequently, the cells were harvested 12 h later, and the activities of firefly luciferase and Renilla luciferase were measured using the dual-luciferase reporter assay system (Promega, Madison, WI, USA), following the manufacturer’s instructions. The obtained data were presented as the relative firefly luciferase activity normalized to Renilla luciferase activity, based on the results of three independently conducted experiments.

### 2.5. ELISA Assay for IFN-β Protein

IPI-FX cells were transfected with the GDS04 nsp5 expression plasmid in 24-well plates. After 24 h of transfection, the cells were either treated with poly(I:C) or left untreated for an additional 16 h. Subsequently, the culture supernatants were collected and subjected to ELISA assay using a porcine IFN-β detection kit (Cusabio, Wuhan, China), following the manufacturer’s instructions.

### 2.6. Indirect Immunofluorescence Assay (IFA)

IPI-FX cells were infected with GDS04 at a MOI of 1 with 10 μg/mL trypsin. Cells treated with trypsin alone (without GDS04 inoculation) were set as the negative control. At 24 and 36 h post-infection (hpi), the cells were fixed with 4% paraformaldehyde for 10 min. After three washes with PBS, the cells were permeabilized with 0.1% Triton X-100 for 10 min at room temperature. Following three additional washes with PBS, the cells were blocked with PBS containing 5% bovine serum albumin (BSA) for 1 h and then incubated with a mouse polyclonal antibody against GDS04 N protein (1:200) for 1 h at room temperature. After three washes with PBS, the cells were stained with an FITC-conjugated secondary antibody for 1 h, followed by DAPI staining for 15 min at room temperature. After three final washes with PBS, the cells were observed under a fluorescence microscope (NIKON Eclipse 80i, Nikon, Tokyo, Japan) to capture fluorescent images.

### 2.7. Extraction of Nuclear and Cytoplasmic Proteins

IPI-FX cells were seeded in 6-well plates and transfected with the GDS04 nsp5 expression plasmid. After 24 h, the cells were either treated with poly(I:C) or left untreated for an additional 12 h. Subsequently, the nuclear and cytoplasmic proteins were extracted using the NE-PER nuclear and cytoplasmic extraction reagents (Thermo Scientific, Waltham, MA, USA), following the manufacturer’s protocol.

### 2.8. Western Blot Analysis

IPI-FX cells grown in 6-well plates were lysed in RIPA Lysis Buffer supplemented with a cocktail of protease and phosphatase inhibitors. The protein expression in each sample was analyzed by SDS-PAGE and transferred onto a polyvinylidene difluoride (PVDF) immunoblotting membrane (Millipore, Billerica, MA, USA). After blocking with 5% milk in 1× TBST, the membrane was incubated with the appropriate primary antibody at room temperature for 1 h, followed by incubation with the corresponding secondary antibody for 1 h at room temperature. The expression of β-actin or GAPDH was detected as a loading control.

### 2.9. Co-Immunoprecipitation Analysis

IPI-FX cells grown in 60 mm dishes were transfected with the expression plasmids, and the whole-cell lysates were harvested using a lysis buffer (Beyotime, Shanghai, China) for Western blot analysis and immunoprecipitation. The cell lysates were incubated overnight at 4 °C with the specific antibody, followed by immunoprecipitation using protein A + G agarose beads (Beyotime, Shanghai, China). After five washes with the lysis buffer, the protein A + G agarose beads were mixed with 50 μL of 1× SDS-PAGE Sample Loading Buffer (Beyotime, Shanghai, China) and boiled for 5 min at 100 °C. The protein expression after immunoprecipitation was then analyzed by Western blot.

### 2.10. Statistical Analysis

All data were presented as the mean ± standard deviation (SD) from independent experiments performed in triplicate. Analyses of significant difference were performed using Student’s *t* test, and *p* values of less than 0.05 were considered statistically significant.

## 3. Results

### 3.1. SADS-CoV GDS04 Suppresses Poly(I:C)-Induced IFN-β Production in IPI-FX Cells

IPI-FX cells, a porcine ileum epithelial cell line, are highly susceptible to SADS-CoV [35]. Considering the differences in gene sequence and pathogenicity between the SADS-CoV GDS04 strain and the SADS-CoV/CN/GDGL/2017 strain, which suppress IFN-β production in IPEC-J2 cells [30,36], we aimed to investigate whether the GDS04 strain could also inhibit IFN-β production in IPI-FX cells. Initially, the replication characteristics of the GDS04 strain were examined in IPI-FX cells. The cells were infected with GDS04 at an MOI of 1, and RT-qPCR analysis revealed a rapid increase in GDS04 N gene mRNA levels at 12 hpi, reaching the highest level at 48 hpi (Figure 1A). Consistent with the RT-qPCR results, Western blot analysis of GDS04 N protein expression showed a similar growth trend over the course of infection (Figure 1B). IFA using a polyclonal antibody against GDS04 N protein at 24 and 36 hpi demonstrated a large number of fluorescence signals coinciding with cell detachment in the virus infection group, while no signals and cytopathic effect were observed in the control group (Figure 1C). Collectively, these findings indicate that SADS-CoV GDS04 can successfully infect IPI-FX cells.

To investigate whether GDS04 has the ability to inhibit activation of the IFN-β promoter, IPI-FX cells cultured in 24-well plates were co-transfected with IFN-β-Luc and phRL-TK, and then infected with GDS04 at an MOI of 1 and transfected with poly(I:C) (1 μg/well) at 24 hpi. After 12 h, the relative activity of the IFN-β promoter was detected by the luciferase reporter gene assay. As depicted in Figure 2A, transfection with poly(I:C) as a positive control strongly induced activation of the IFN-β promoter in uninfected cells. However, in GDS04-infected cells, the promoter activities of IFN-β induced by poly(I:C) were significantly inhibited (*p* < 0.01). Furthermore, GDS04 also inhibited both the mRNA expression and protein level of IFN-β induced by poly(I:C), as detected by RT-qPCR and ELISA (Figure 2B,C). Additionally, the induction of IFN-β requires the activation of transcription factors IRF3 and NF-κB. As illustrated in Figure 2D,E, GDS04 infection inhibited the promoter activities of IRF3 and NF-κB induced by poly(I:C). Taken together, these results indicate that GDS04 infection suppressed poly(I:C)-induced IFN-β production by inhibiting the promoter activities of IRF3 and NF-κB.

### 3.2. SADS-CoV nsp5 Is Identified as an Interferon Inhibitor

SADS-CoV encodes 23 proteins, including 16 non-structural proteins, four structural proteins, and three accessory proteins [1]. Coronavirus nsp5 is a cysteine proteinase, whose catalytic domain exhibits a high degree of conservation, including the presence of histidine at position 41 and cysteine at position 144 [17]. To investigate the potential of SADS-CoV nsp5 in suppressing IFN-β production, IPI-FX cells were transfected with poly(I:C) 24 h after transfection with the nsp5 expression plasmid, and the activation of the IFN-β promoter was assessed using luciferase reporter gene assay. As depicted in Figure 3A, the expression of SADS-CoV nsp5 effectively inhibited poly(I:C)-induced IFN-β promoter activation. To further confirm the inhibitory role of nsp5 against IFN-β, the mRNA expression and protein level of IFN-β were evaluated. As illustrated in Figure 3B,C, SADS-CoV nsp5 significantly suppressed poly(I:C)-induced IFN-β synthesis (*p* < 0.0001). Moreover, the inhibitory effect of SADS-CoV nsp5 on IFN-β production induced by poly(I:C) was observed in a dose-dependent manner (Figure 3B). Collectively, these findings demonstrate that SADS-CoV nsp5 functioned as an IFN inhibitor, suppressing the poly(I:C)-induced IFN-β production.

### 3.3. SADS-CoV nsp5 Exhibits Inhibitory Effects on Poly(I:C)-Induced Activation of IRF3

To elucidate the underlying mechanism by which SADS-CoV nsp5 inhibits IFN-β induction in IPI-FX cells, the promoter activities of IRF3 and NF-κB, induced by poly(I:C), were assessed following nsp5 overexpression. IPI-FX cells were co-transfected with IRF3-Luc/NF-κB-Luc, along with phRL-TK and the nsp5 expression plasmid, and subsequently transfected with poly(I:C). The relative activity of the IRF-3 and NF-κB promoters was determined using a dual-luciferase assay. As depicted in Figure 4A,B, the promoter activities of both IRF3 (*p* < 0.001) and NF-κB (*p* < 0.05), induced by poly(I:C), were significantly suppressed by nsp5.

Given that IRF3 serves as a crucial transcription factor for IFN-β production, its activation necessitates phosphorylation and nuclear translocation. To explore the impact of SADS-CoV nsp5 on the phosphorylation and nuclear translocation of IRF3, IPI-FX cells expressing SADS-CoV nsp5 were transfected with poly(I:C). Western blot analysis of the cell extracts revealed a significant reduction in IRF3 phosphorylation in the nsp5-expressing cells (Figure 4C). Furthermore, analysis of nuclear and cytoplasmic protein fractions demonstrated a notable inhibition of IRF3 nuclear translocation by nsp5 protein (Figure 4D). Collectively, these results indicate that SADS-CoV nsp5 hampered the activation of IRF3, thereby exerting inhibitory effects on IFN-β production.

### 3.4. SADS-CoV nsp5 Hinders the Induction of IFN-β Mediated by IKKε

Given the previously observed inhibition of IRF3 activation by SADS-CoV nsp5, there is a possibility that nsp5 may target IRF3 to suppress IFN-β production. To investigate this hypothesis, the expression plasmids of IRF3 and nsp5 were co-transfected into IPI-FX cells, and the levels of IFN-β mRNA were assessed. Figure 5A demonstrated that SADS-CoV nsp5 did not impede IFN-β production induced by IRF3, indicating that nsp5 likely targets upstream components of the IRF3 signaling pathway involved in IFN-β production. To identify the potential target of SADS-CoV nsp5, TBK1 and IKKε, two essential kinases involved in IRF3 activation, were examined. The expression of IKKε and TBK1 was initially confirmed through Western blot analysis. Subsequently, IPI-FX cells were co-transfected with expression plasmids of IKKε or TBK1 along with nsp5. As depicted in Figure 5B,C, SADS-CoV nsp5 significantly inhibited INF-β production induced by IKKε (*p* < 0.001), while it failed to impede TBK1-induced IFN-β production. These findings suggest that SADS-CoV nsp5 likely targeted IKKε.

To further confirm the potential interaction between SADS-CoV nsp5 and IKKε, IPI-FX cells were co-transfected with Myc-nsp5 and HA-IKKε expression plasmids. After 48 h, whole-cell lysates were collected for an immunoprecipitation assay. The results in Figure 5D,E demonstrated the detection of Myc-nsp5 or HA-IKKε proteins in the precipitant of HA-IKKε or Myc-nsp5, respectively, indicating a direct interaction between SADS-CoV nsp5 and IKKε. These results collectively suggest that SADS-CoV nsp5 impeded IFN-β induction by targeting IKKε.

### 3.5. SADS-CoV nsp5 Diminishes the Protein Abundance of IKKε and Hinders Its Phosphorylation

To further elucidate the mechanism by which SADS-CoV nsp5 affects IKKε, the protein abundance of IKKε was assessed in IPI-FX cells transfected with the nsp5 expression plasmid. Figure 6A demonstrates a significant reduction in IKKε protein abundance in nsp5-overexpressing IPI-FX cells. Additionally, the phosphorylation of IKKε at the Ser172 position was examined, and the results in Figure 6B reveal that SADS-CoV nsp5 inhibited the phosphorylation of IKKε. Furthermore, IPI-FX cells were either uninfected or infected with SADS-CoV GDS04 for 12, 24, and 36 h, and the expression level of IKKε protein was analyzed by Western blot. As depicted in Figure 6C, the prolonged infection with SADS-CoV GDS04 led to a gradual decrease in the protein abundance of endogenous IKKε, providing evidence that SADS-CoV nsp5 reduced the protein abundance of IKKε. These results collectively indicate that SADS-CoV nsp5 interacted with IKKε, resulting in reduced IKKε protein abundance and inhibition of its phosphorylation.

## 4. Discussion

Coronaviruses, including SADS-CoV, can cause respiratory or gastrointestinal diseases in humans and animals. To successfully replicate and proliferate within host cells while evading the host’s antiviral response, coronaviruses have evolved various strategies to counteract IFN-β production. The strategies involve regulating phosphorylation or ubiquitination-related pathways, cleaving important signal molecules in the IFN-β production pathway, or modulating the transcription and replication of host genes [37]. Previous studies have reported that coronaviruses such as SADS-CoV [30,31,36], PEDV [14], PDCoV [38], SARS-CoV [39], SARS-CoV-2 [12,13], MERS-CoV [40], and their encoded proteins can inhibit IFN-β production. For instance, the N proteins of PDCoV and SADS-CoV mediate K63-linked ubiquitination of RIG-I, thereby inhibiting host IFN-β production. In our study, we observed that SADS-CoV nsp5 protein inhibited poly(I:C)-induced IFN-β production by inhibiting the IRF3 signaling pathway in IPI-FX cells (Figure 7).

Coronavirus nsp5 encodes a 3C-like protease with cysteine protease activity that plays a crucial role in virus replication and transcription. Previous studies on coronaviruses such as PEDV, PDCoV, SARS-CoV, SARS-CoV-2, and MERS-CoV have demonstrated that nsp5 can inhibit IFN-β and has been considered an important target of drug treatment against single-stranded positive-strand RNA viruses [15,16]. PEDV and PDCoV nsp5 can cleave NEMO, a regulator of the transcription factor NF-κB, by cleaving glutamine at position 231 [23,41]. Additionally, PDCoV nsp5 cleaves signal transducer and activator of transcription 2 (STAT2) to inhibit the interferon response pathway [29], and cleaves interferon-stimulating gene DCP1A to reduce its antiviral activity [42]. PEDV has also been reported to inhibit the interferon response pathway by cleaving STAT1 [43].

Recently, SADS-CoV nsp5 was reported to suppress type I interferon signaling by cleaving DCP1A [44]. DCP1A, as an upstream component of IRF3, suppresses the IRF3 and NF-κB signaling pathways, thereby reducing the production of IFN-β and inflammatory cytokines. Additionally, in that study, overexpression of SADS-CoV nsp5 decreased the expression of IKKα, IKKβ, and IκBα (which regulate the NF-κB signaling pathway). However, that study did not involve the IKKε protein (which regulates IRF3 signaling). In our study, we discovered that SADS-CoV nsp5 interacted with a novel signaling molecule, IKKε, which forms a complex with TBK1 to activate IRF3 and promote IFN-β production. Furthermore, we observed that SADS-CoV nsp5 reduced the endogenous protein level of IKKε and inhibited its phosphorylation, leading to impaired IRF3 activation and nuclear translocation. The results of the two studies are consistent in that they both demonstrate that nsp5 suppresses IFN-β production by acting on upstream molecules of IRF3. Consequently, IFN-β production is diminished, allowing SADS-CoV to evade the host’s innate immunity and thus enhancing the pathogenicity after viral infection. However, the above-mentioned studies were all conducted based on in vitro cell lines. Given the complexity of animal body structures and physiological conditions, the contribution of SADS-CoV nsp5 protein to immunosuppressive function during viral infection in animals and the core mechanisms by which it exerts immunosuppression still need to be further explored.

## 5. Conclusions

The innate immune system serves as the initial defense mechanism against pathogens, and the production of type I IFNs plays a critical role in the antiviral response by establishing an antiviral state in both infected and uninfected cells. In this study, we discovered that SADS-CoV nsp5 interfered with IFN-β production through its interaction with IKKε, resulting in a reduction in endogenous IKKε protein levels and inhibiting its phosphorylation activation. Consequently, the activation of the IRF3 signaling pathway, which is essential for IFN-β production, was suppressed. These findings contribute to a better understanding of the immune evasion mechanisms employed by coronaviruses and provide valuable insights for future investigations into the role of coronaviral nsp5 in immune escape.

## Figures and Tables

**Figure 1 microorganisms-13-01494-f001:**
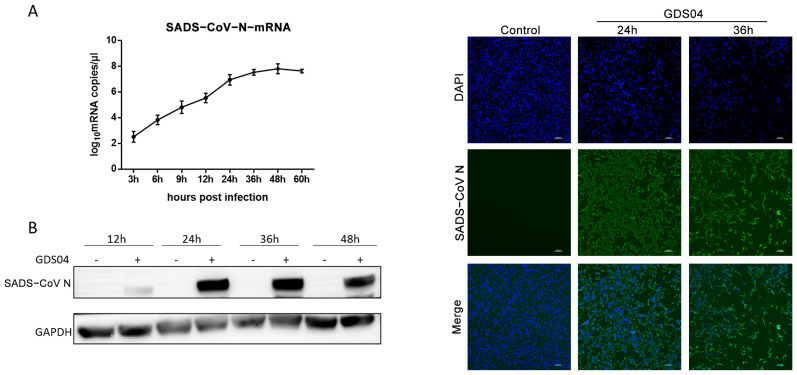
The replication characteristics of SADS-CoV GDS04 in IPI-FX cells. (**A**) Expression of SADS-CoV N mRNA in infected cells detected by RT-qPCR. IPI-FX cells were infected with GDS04 (MOI = 1). The mRNA levels of the N gene were determined at 3, 6, 9, 12, 24, 36, 48, and 60 hpi using RT-qPCR. Cells treated with trypsin alone (without GDS04 inoculation) were set as the negative control. The results are represented as the mean ± SD with three replicates. (**B**) Expression of SADS-CoV N protein in infected cells detected by Western blot analysis. IPI-FX cells were infected with GDS04 (MOI = 1). At 12, 24, 36, and 48 hpi, cell extracts were prepared and subjected to Western blot analysis. (**C**) Expression of SADS-CoV N protein in infected cells detected by immunofluorescence assay. IPI-FX cells were infected with GDS04 (MOI = 1). Cells treated with trypsin alone (without GDS04 inoculation) were set as the negative control. At 24 and 36 hpi, the cells were fixed and incubated with a polyclonal antibody against SADS-CoV N protein (green). The resulting images were captured using a fluorescence microscope (Bar = 100 μm).

**Figure 2 microorganisms-13-01494-f002:**
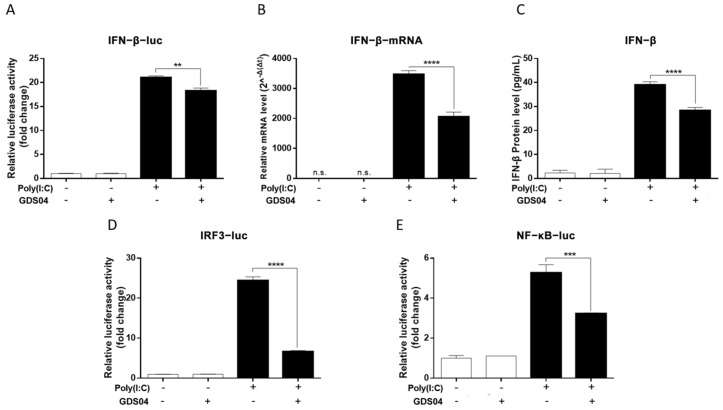
SADS-CoV GDS04 inhibits poly(I:C)-induced IFN-β production and promoter activation. (**A**) IPI-FX cells were co-transfected with IFN-β-Luc and phRL-TK, and then infected with GDS04 (MOI = 1) for 24 h. Subsequently, the cells were either transfected or not transfected with poly(I:C) for an addition 12 h. The relative activity of the IFN-β promoter was determined using a dual-luciferase assay. (**B**) IPI-FX cells were either uninfected or infected with GDS04 (MOI = 1) for 24 h. Subsequently, the cells were transfected or not transfected with poly(I:C) for an additional 12 h. Total RNA was extracted to determine the relative mRNA expression of IFN-β by RT-qPCR. (**C**) IPI-FX cells were either uninfected or infected with GDS04 (MOI = 1) for 24 h, followed by transfection or no transfection with poly(I:C) for an additional 12 h. The supernatants were then collected for an ELISA analysis with a porcine IFN-β detection kit. (**D**,**E**) IPI-FX cells were co-transfected with IRF3-Luc (**D**)/NF-κB-Luc (**E**) and phRL-TK, and then infected with GDS04 (MOI = 1) for 24 h. Subsequently, the cells were either transfected or not transfected with poly(I:C) for an additional 12 h. The relative activity of the IRF-3 and NF-κB promoters was determined using a dual-luciferase assay. All data are represented as the mean  ±  SD from three replicates. **, *p*  <  0.01; ***, *p*  <  0.001; ****, *p*  <  0.0001.

**Figure 3 microorganisms-13-01494-f003:**
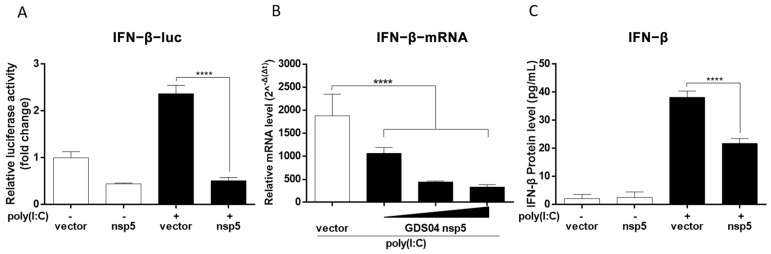
SADS-CoV nsp5 inhibits IFN-β production. (**A**) IPI-FX cells were co-transfected with IFN-β-Luc and phRL-TK, along with either a vector or nsp5 expression plasmid, for 24 h. Subsequently, the cells were either transfected or not transfected with poly(I:C) for an additional 12 h. The relative activity of the IFN-β promoter was assessed using a dual-luciferase assay. (**B**) IPI-FX cells were transfected with 0.2/0.4/0.8 μg of the nsp5 expression plasmid, respectively, and transfected with poly(I:C) 36 h later. After 12 h, total RNA was extracted, and the relative mRNA expression of IFN-β was determined using RT-qPCR. (**C**) IPI-FX cells were transfected with either a vector or nsp5 expression plasmid for 24 h. Then, the cells were either transfected or not transfected with poly(I:C) for an additional 12 h. The supernatants were collected and subjected to ELISA analysis using a porcine IFN-β detection kit. All data are represented as the mean  ±  SD from three independent replicates. ****, *p* < 0.0001.

**Figure 4 microorganisms-13-01494-f004:**
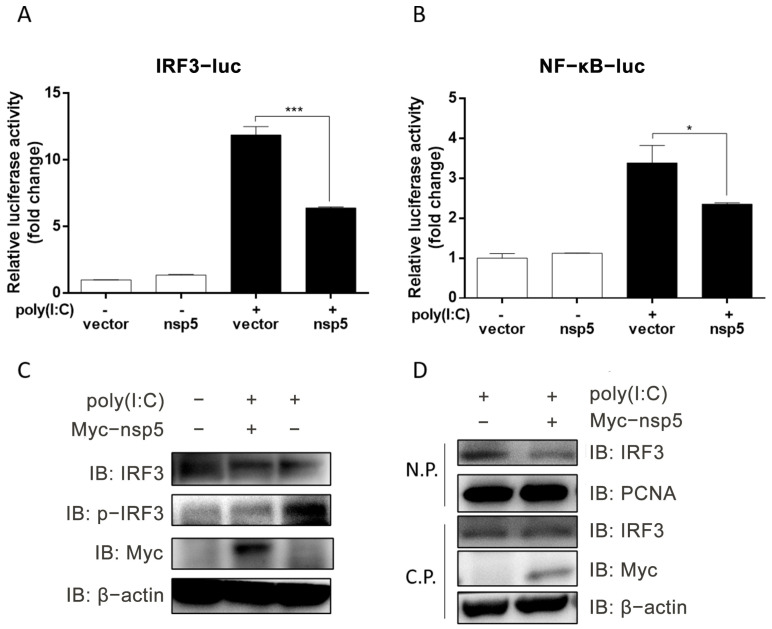
SADS-CoV nsp5 inhibits poly(I:C)-induced activation of IRF3. (**A**,**B**) IPI-FX cells were co-transfected with IRF3-Luc (**A**) NF-κB-Luc (**B**), and phRL-TK, along with either a vector or nsp5 expression plasmid for 36 h. Subsequently, the cells were transfected or not transfected with poly(I:C) for an additional 12 h. The relative activity of the IRF-3 and NF-κB promoters was determined using a dual-luciferase assay. (**C**,**D**) IPI-FX cells were transfected with either a vector or nsp5 expression plasmid for 36 h, followed by transfection or not with poly(I:C) for an additional 12 h. Cell extracts (**C**) or nuclear and cytoplasmic proteins (**D**) were prepared and subjected to Western blot analysis. N.P., nuclear protein; C.P., cytoplasmic protein. All data are represented as the mean  ±  SD from three independent replicates. *, *p* < 0.05; ***, *p* < 0.001.

**Figure 5 microorganisms-13-01494-f005:**
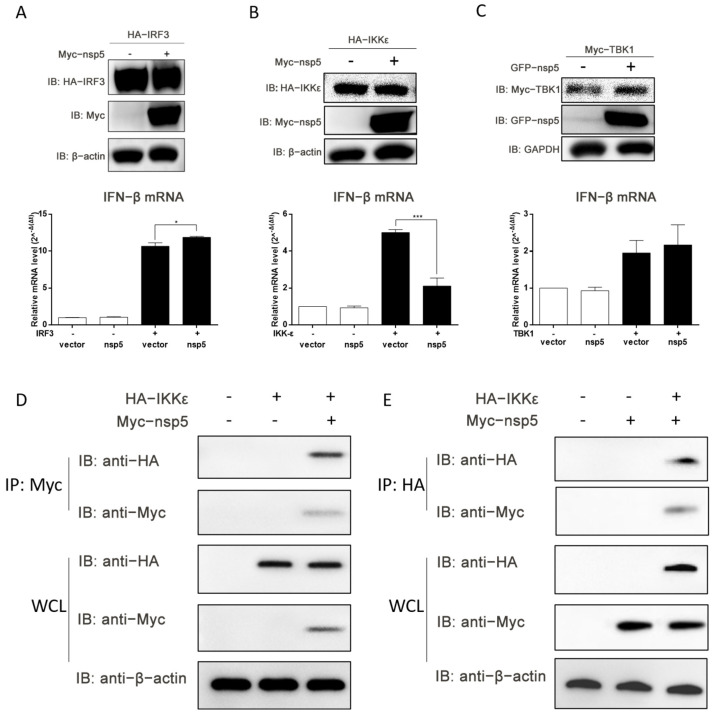
SADS-CoV nsp5 impedes IFN-β induction mediated by IKKε. (**A**–**C**) IPI-FX cells were co-transfected with expression plasmids of SADS-CoV nsp5 and IRF3 (**A**), IKKε (**B**), or TBK1 (**C**) for 48 h. The relative mRNA levels of IFN-β were determined using RT-qPCR. Cell extracts were prepared and subjected to Western blot analysis. (**D**,**E**) IPI-FX cells were co-transfected with expression plasmids of HA-IKKε and Myc-nsp5. After 48 h, whole-cell lysates were collected for an immunoprecipitation assay, and the protein expression levels were analyzed by Western blot. All data are represented as the mean ± SD from three independent replicates. *, *p* < 0.05; ***, *p* < 0.001.

**Figure 6 microorganisms-13-01494-f006:**
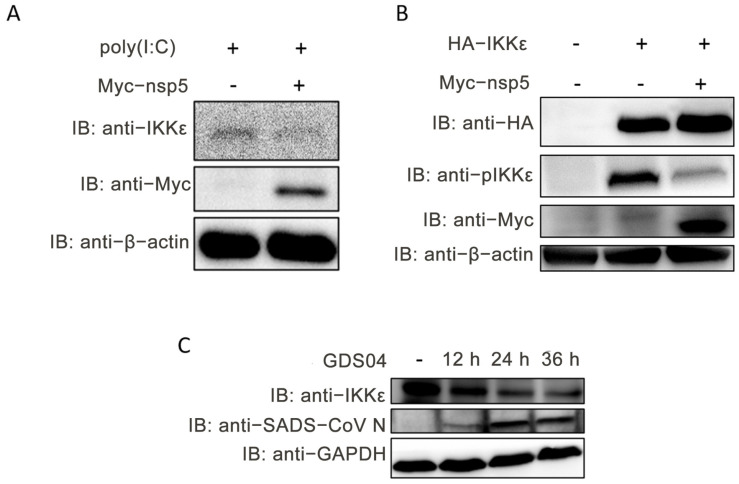
SADS-CoV nsp5 reduces the abundance of IKKε protein and inhibits its phosphorylation. (**A**) IPI-FX cells were transfected with either a vector or the nsp5 expression plasmid for 36 h, followed by transfection with or without poly(I:C) for an additional 12 h. Cell extracts were prepared and subjected to Western blot analysis. (**B**) IPI-FX cells were co-transfected with either a vector or the nsp5 expression plasmid along with HA-IKKε for 48 h. Cell extracts were prepared and subjected to Western blot analysis. (**C**) IPI-FX cells were either uninfected or infected with GDS04 (MOI = 1). At 12, 24, and 36 hpi, cell extracts were prepared and subjected to Western blot analysis.

**Figure 7 microorganisms-13-01494-f007:**
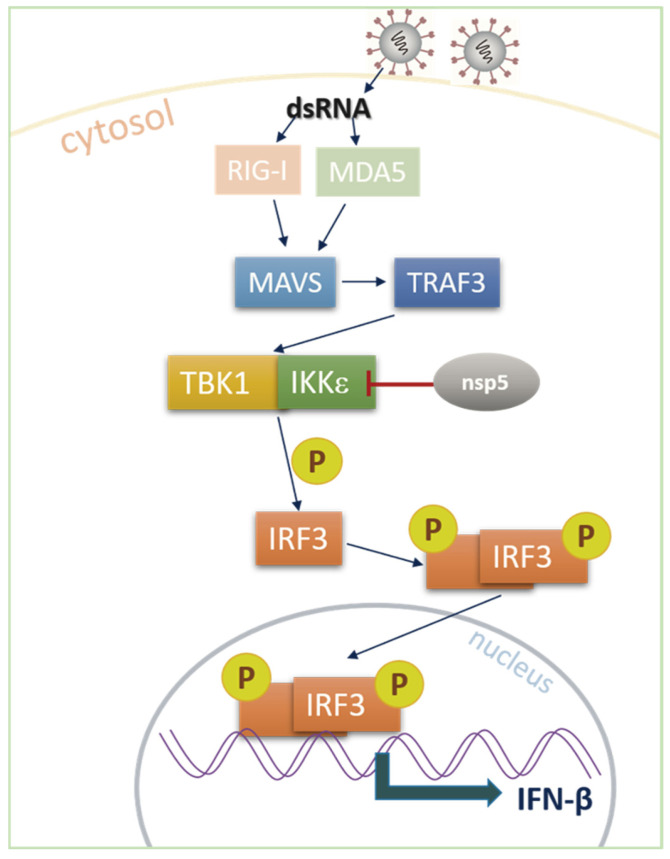
Schematic diagram of SADS-CoV nsp5 inhibiting interferon production by targeting kinase IKKε in this study.

**Table 1 microorganisms-13-01494-t001:** Primers used in qPCR.

Primers	Sequence (5′-3′)	Position
SADS-CoV-N-qPCR-F	CTGACTGTTGTTGAGGTTAC	415 bp–434 bp
SADS-CoV-N-qPCR-R	TCTGCCAAAGCTTGTTTAAC	550 bp–569 bp
IFN-β-qPCR-F	AGTGCATCCTCCAAATCGCT	11 bp–30 bp
IFN-β-qPCR-R	GCTCATGGAAAGAGCTGTGGT	49 bp–69 bp
GAPDH-qPCR-F	CCTTCCGTGTCCCTACTGCCAAC	901 bp–923 bp
GAPDH-qPCR-R	GACGCCTGCTTCACCACCTTCT	982 bp–1003 bp

## Data Availability

The data presented in this study are openly available in [FigShare] at [https://www.doi.org/10.6084/m9.figshare.28980455].

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
