# Peer review of "SADS-CoV nsp5 Inhibits Interferon Production by Targeting Kinase IKKε"

_microorganisms, 2025, doi:10.3390/microorganisms13071494_

Round 1
Reviewer 1 Report
Comments and Suggestions for Authors
The topic is relevant - given the main problem in the fight against viral infections, namely the escape from the immune response through various mechanisms - antigenic variations, molecular mimicry, insufficient neutralizing antibodies, etc.
I have recommendations and remarks for the authors, some of which are serious.
1. The overall impression I had was not particularly scientific in its style of expression. The style is somewhat like that of a popular science magazine. The word as "antagonize" seems extremely inappropriate to me. It would be better to suppress, inhibit, or mimics or something similar. It is not clear."Collectively, these findings indicate efficient proliferation of SADS-CoV GDS04 in IPI-FX cells at 24 hpi"-this sentence on line 170 is extremely incorrect, unscientific, and unvirological. This is probably a case of complete virus-induced cell disintegration or a viral cytopathic effect affecting the entire monolayer, etc.
2. I recommend that the authors explain in more detail the function of nsp5 as well as the mechanisms by which it inhibits IFN-β /or inhibition of the interferon receptor/, NF-κB promoters- It inhibits synthesis, signal transduction, synthesizes the so-called virokines (soluble molecules that mimic cytokines) in the introduction section. I think a figure, a diagram, a picture would help you. When you write in the discussion that you have discovered a third way to inhibit interferon, it is not bad to explain the other two.
3. Materials and methods section - not described clearly enough. For example, how long are the cell cultures incubated, are you working with a dense monolayer, is the culture a monolayer at all. It is not clear whether you constructed the plasmid expressing nsp5. You should indicate the positions of the primers in table 1. It's not clear to me why you are looking for a fragment of the N protein gene. How long is this fragment and why are you looking for it in the first place?
4. Line 173" IPI-FX cells were either mock-infected or infected with GDS04 (MOI=1)"-What do you mean, what does mock-infected mean? Do you mean cellular control? The photo in Figure 1 is of very poor quality and nothing is visible. Is there a control for the experiment?
Please check the figure numbers. Figure 2 is missing. You could generally make the figures clearer, as the inscriptions are difficult to see.
4. Line 165-You claim that the levels of mRNA for translation of the N protein reach the highest levels at 48 hours after infection. Has the Ct been calculated, which is quite important?
5. Line 170-please don't talk about proliferation and viruses. It's unacceptable. In virology, it's used-virus replication, virus replication cycle. The same applies to "mock-infected" cells. It is probably a control cell that is not infected with the virus or one that has only been transfected/why the term is transfection/ with poly(I:C) . It is not clear.
6. Line 218-" IPI-FX cells expressing SADS-CoV nsp5
protein"-sorry what are these cells, how exactly were they constructed? If this is true, please describe in the materials and methods section.
7. Line 228- "virus-encoded IFN antagonist"-please explain how exactly. Does the protein itself mimic IFN-β, does it preferentially bind to an IFN-β receptor or does it affect the IFN-β promoter by a cis or trans mechanism or in some other way.
8. Which of all the mechanisms described in sections 3.3, 3.4, 3.5 is new and unknown in your opinion? It might be a good idea to note which is your contribution.
9. Line 327 - this conclusion is important and I recommend that it be in the conclusion.
10. The conclusion on lines 330-333, is it your merit or are there studies before this, since there are many such studies on viral proteins interfering with cellular macromolecules, including for SARS-CoV-2? Where is Figure 7 taken from? Please check.
11. Lines 347-349 you yourself mention that this non-structural protein has such a role in other types of coronaviruses of the betacoronavirus genus and this has already been proven. Do I understand that your contribution is that you prove the same thing, but for another representative of this family?
In the conclusion section, I recommend that the authors specify what exactly is new and unexplored that they prove. Also, how exactly did you approach nsp5 if it is not previous research and publications. I think this should be clarified in the conclusion.
Author Response
- The overall impression I had was not particularly scientific in its style of expression. The style is somewhat like that of a popular science magazine. The word as "antagonize" seems extremely inappropriate to me. It would be better to suppress, inhibit, or mimics or something similar. It is not clear."Collectively, these findings indicate efficient proliferation of SADS-CoV GDS04 in IPI-FX cells at 24 hpi"-this sentence on line 170 is extremely incorrect, unscientific, and unvirological. This is probably a case of complete virus-induced cell disintegration or a viral cytopathic effect affecting the entire monolayer, etc.
Response 1: Thank you for pointing this out. First, in accordance with your suggestion, we have replaced “antagonize” in the manuscript with “inhibit” and “suppress”. Second, we have revised the sentence in Line 170 to “Collectively, these findings indicate that SADS-CoV GDS04 can successfully infect IPI-FX cells.” The corresponding changes have been highlighted in the revised manuscript.
- I recommend that the authors explain in more detail the function of nsp5 as well as the mechanisms by which it inhibits IFN-β /or inhibition of the interferon receptor/, NF-κB promoters- It inhibits synthesis, signal transduction, synthesizes the so-called virokines (soluble molecules that mimic cytokines) in the introduction section. I think a figure, a diagram, a picture would help you. When you write in the discussion that you have discovered a third way to inhibit interferon, it is not bad to explain the other two.
Response 2: Thank you for pointing this out. And we have added relevant content to the introduction section.
- Materials and methods section - not described clearly enough. For example, how long are the cell cultures incubated, are you working with a dense monolayer, is the culture a monolayer at all. It is not clear whether you constructed the plasmid expressing nsp5. You should indicate the positions of the primers in table 1. It's not clear to me why you are looking for a fragment of the N protein gene. How long is this fragment and why are you looking for it in the first place?
Response 3: Thank you for pointing this out. Cells in this study were cultured and performed as confluent monolayers. We have added this in revised-manuscript in the “2.1 virus, cells and reagents” section, and highlighted it. The expression plasmids of SADS-CoV nsp5 were constructed and described in the "2.2 Plasmids" section (Line 88) of the manuscript. The amplification of the N protein fragment serves to determine viral proliferation. The coronavirus N protein, as the most stable and conserved structural protein, is routinely used as a viral detection marker.
- Line 173" IPI-FX cells were either mock-infected or infected with GDS04 (MOI=1)"-What do you mean, what does mock-infected mean? Do you mean cellular control? The photo in Figure 1 is of very poor quality and nothing is visible. Is there a control for the experiment?
Please check the figure numbers. Figure 2 is missing. You could generally make the figures clearer, as the inscriptions are difficult to see.
Response 4: Thank you for pointing this out. The “mock-infected” means cells were treated with trypsin along without SADS-CoV inoculation. In the revised manuscript and figures, we have revised the relevant sentences and supplemented the control group information. In Figure 1C, we have increased the brightness of the images to enhance visibility. There is a control group in all the experiments, and we have re-noted it in figures.
In the original manuscript we submitted, an error occurred where Figure 2 was incorrectly labeled as Figure 1, resulting in the missing of Figure 2. This mistake has been corrected in the revised manuscript and highlighted it.
- Line 165-You claim that the levels of mRNA for translation of the N protein reach the highest levels at 48 hours after infection. Has the Ct been calculated, which is quite important?
Response 5: Thank you for pointing this out. The mRNA expression levels of N protein were calculated via a standard curve (Ct values vs. molecular copy numbers) generated using standard plasmids. Consequently, Ct values are not explicitly presented in the text or Figure 1A. This result shown was derived from these Ct values and the corresponding standard curve.
- Line 170-please don't talk about proliferation and viruses. It's unacceptable. In virology, it's used-virus replication, virus replication cycle. The same applies to "mock-infected" cells. It is probably a control cell that is not infected with the virus or one that has only been transfected/why the term is transfection/ with poly(I:C) . It is not clear.
Response 6: Thank you for pointing this out. We have revised “proliferation” to “replication” and highlighted it in revised manuscript. The “mock-infected” means cells were treated with trypsin along without SADS-CoV inoculation. In the revised manuscript and figures, we have revised the relevant sentences and supplemented the control group information.
- Line 218-" IPI-FX cells expressing SADS-CoV nsp5 protein"-sorry what are these cells, how exactly were they constructed? If this is true, please describe in the materials and methods section.
Response 7: Thank you for pointing this out. “IPI-FX cells expressing SADS-CoV nsp5 protein” means cells were transfected with nsp5 expression plasmid. We have revised this sentence in revised manuscript and highlighted it.
- Line 228- "virus-encoded IFN antagonist"-please explain how exactly. Does the protein itself mimic IFN-β, does it preferentially bind to an IFN-β receptor or does it affect the IFN-β promoter by a cis or trans mechanism or in some other way.
Response 8: Thank you for pointing this out. In this section, we only found that nsp5 acts as an IFN inhibitor, suppressing the production of IFN-β induced by poly(I:C). However, the mechanism by which nsp5 exerts its IFN inhibitory effect will be investigated in the subsequent part (sections 3.3, 3.4, 3.5).
- Which of all the mechanisms described in sections 3.3, 3.4, 3.5 is new and unknown in your opinion? It might be a good idea to note which is your contribution.
Response 9: Thank you for pointing this out. The section 3.4 and 3.5 is new.
- Line 327 - this conclusion is important and I recommend that it be in the conclusion.
Response 10: We agree with this comment. Therefore, we adjusted the position of this section to the “conclusion” section.
- The conclusion on lines 330-333, is it your merit or are there studies before this, since there are many such studies on viral proteins interfering with cellular macromolecules, including for SARS-CoV-2? Where is Figure 7 taken from? Please check.
Response 11: Thank you for pointing this out. The studies described in lines 330-333 are previously reported works and not our merit. Relevant references have been added accordingly. Figure 7 is a self-drawn mechanism summary of nsp5-mediated IFN inhibition in this study, and its citation is embedded within the conclusion sentences of our research.
- Lines 347-349 you yourself mention that this non-structural protein has such a role in other types of coronaviruses of the betacoronavirus genus and this has already been proven. Do I understand that your contribution is that you prove the same thing, but for another representative of this family?
Response 12: Thank you for your comment. We did demonstrate the IFN inhibitory effect of nsp5 in another representative of this family, but the key distinction here is that the mechanism by which SADS-CoV nsp5 suppresses IFN differs from previously published reports, which represents a major highlight of this study.
Reviewer 2 Report
Comments and Suggestions for Authors
The manuscript by She et al. describes the interferon-inhibitory role of the nsp5 protein of the swine acute diarrhea syndrome coronavirus (SADS-CoV).
All experiments were conducted in vitro using the porcine ileum epithelial cell line. The authors have demonstrated the pathway of inhibition of interferon production through the interaction of nsp5 and IKKε.
Figure 7 shows the schematic of the mechanism of dsRNA inducing interferon production and how nsp5 inhibits the cascade. This is great for any reader to understand the mechanism.
Author Response
The manuscript by She et al. describes the interferon-inhibitory role of the nsp5 protein of the swine acute diarrhea syndrome coronavirus (SADS-CoV).
All experiments were conducted in vitro using the porcine ileum epithelial cell line. The authors have demonstrated the pathway of inhibition of interferon production through the interaction of nsp5 and IKKε.
Figure 7 shows the schematic of the mechanism of dsRNA inducing interferon production and how nsp5 inhibits the cascade. This is great for any reader to understand the mechanism.
Response: Thank you for your recognition of our work.
Reviewer 3 Report
Comments and Suggestions for Authors
Abstract
Clarity: While the abstract presents a logical progression, the writing is dense and could be streamlined. Consider simplifying sentence structures for better readability.
Introduction
Flow: Transition from general coronavirus mechanisms to specific SADS-CoV functions is abrupt. Suggest smoother transitions and clearer hypothesis development.
Materials and Methods
Detailing: The description of reagents and protocols is mostly sufficient, but key parameters like transfection efficiency, MOI rationale, and antibody validation are missing or underexplained.
Results
3.1 SADS-CoV GDS04 suppresses IFN-β production
Quantitative rigor: Figure legends and results rely heavily on phrases like “significantly inhibited” without reporting actual p-values or fold-changes.
3.2–3.3 nsp5 function and mechanism
Mechanistic gap: While nsp5 is shown to inhibit IRF3 and NF-κB, the causal link to IKKε isn’t clearly bridged until later. Consider restructuring to enhance logical flow.
3.4–3.5 IKKε targeting
Selectivity: It's shown that nsp5 targets IKKε but not TBK1; this needs further exploration—are other signaling kinases involved?
Discussion
Originality: The novelty of targeting IKKε specifically is mentioned but underdeveloped. Compare more explicitly with previous findings on DCP1A and other pathways.
Depth: The discussion could be enriched by speculating on how nsp5-mediated immune evasion might influence SADS-CoV pathogenesis in vivo.
Limitations: No limitations are acknowledged. This weakens the credibility of the conclusions. Suggested: lack of in vivo validation, cell-line limitations, or absence of mutant/cleavage-inactive nsp5.
Conclusion
Overstated: Phrases like “identified a new target” are strong claims based only on co-IP and expression changes. Recommend softening to “suggests that nsp5 targets...”.
Author Response
Abstract
Clarity: While the abstract presents a logical progression, the writing is dense and could be streamlined. Consider simplifying sentence structures for better readability.
Response:Thank you for your comment. In accordance with your comment, we have revised the language in the abstract section. The revised sentences have been highlighted in the revised manuscript.
Introduction
Flow: Transition from general coronavirus mechanisms to specific SADS-CoV functions is abrupt. Suggest smoother transitions and clearer hypothesis development.
Response:Thank you for your comment. We have rearranged the content order in the Introduction section and added transitional sentences to achieve smoother transitions and clearer hypothesis development.
Materials and Methods
Detailing: The description of reagents and protocols is mostly sufficient, but key parameters like transfection efficiency, MOI rationale, and antibody validation are missing or underexplained.
Response:Thank you for your comment. In the Materials and Methods section, we have supplemented some details. Regarding the issue of transfection efficiency you mentioned, we set up control groups in all experiments, and the statistical analysis of the post-transfection results was conducted relative to these controls. As for the MOI rationale, we used an MOI of 1, whichapproximates natural infection dynamics (single-virus entry per cell), and provides consistent, comparable results across studies under Poisson distribution principles.. Regarding antibody validation, the anti-N protein antibody used in this study was prepared by our team. Its specificity and sensitivity were rigorously verified during the preparation. The remaining antibodies, being commercial products, have already demonstrated well - established specificity.
Results
3.1 SADS-CoV GDS04 suppresses IFN-β production
Quantitative rigor: Figure legends and results rely heavily on phrases like “significantly inhibited” without reporting actual p-values or fold-changes.
Response: Thank you for pointing it out. We have added the p-values in the text.
3.2–3.3 nsp5 function and mechanism
Mechanistic gap: While nsp5 is shown to inhibit IRF3 and NF-κB, the causal link to IKKε isn’t clearly bridged until later. Consider restructuring to enhance logical flow.
Response: Thank you for your comments. In the study of the mechanism by which nsp5 inhibits IFN-β, we traced the process step by step from the end of the signaling pathway to identify the action site of nsp5. The production of IFN requires the phosphorylation and nuclear translocation of IRF3 and NF-κB. Overexpression of nsp5 did not impede IFN-β production induced by IRF3, indicating that nsp5 likely targets up-stream components of the IRF3 signaling pathway involved in IFN-β production. TBK1 and IKKε are key proteases for activating IRF3. In this study, we experimentally verified that nsp5 exerts its function by targeting IKKε. Therefore, we conclude that nsp5 targets IKKε to inhibit the activation of IRF3, thereby suppressing IFN-β production. We believe this logic is coherent and reasonable. You suggested strengthening the logical chain of this part. We sincerely hope you can provide specific revision suggestions so that we can learn and make the necessary amendments.
3.4–3.5 IKKε targeting
Selectivity: It's shown that nsp5 targets IKKε but not TBK1; this needs further exploration—are other signaling kinases involved?
Response: Thank you for your comments. We highly appreciate your suggestion and will conduct in-depth research in accordance with it. However, the results of this additional research will not be included in the current manuscript submission.
Discussion
Originality: The novelty of targeting IKKε specifically is mentioned but underdeveloped. Compare more explicitly with previous findings on DCP1A and other pathways.
Response: Thank you for pointing it out. We have compared and discussed the results and conclusions of the two studies, and added the content to the corresponding positions in the manuscript (highlighted).
Depth: The discussion could be enriched by speculating on how nsp5-mediated immune evasion might influence SADS-CoV pathogenesis in vivo.
Response:Thank you for your comment. We agree with this comment, and have mentioned it in the final of the discussion.
Limitations: No limitations are acknowledged. This weakens the credibility of the conclusions. Suggested: lack of in vivo validation, cell-line limitations, or absence of mutant/cleavage-inactive nsp5.
Response:we agree with this comment, and have added relevant sentences in the final of discussion.
Conclusion
Overstated: Phrases like “identified a new target” are strong claims based only on co-IP and expression changes. Recommend softening to “suggests that nsp5 targets...”.
Response:Thank you for your comment. We have revised the relevant expressions.
Round 2
Reviewer 1 Report
Comments and Suggestions for Authors
- Line 75-79 - I understand that one result is new and the other is confirmatory?
- Regarding the answer to question 4: "The “mock-infected” means cells were treated with trypsin along without SADS-CoV inoculation"-This answer is strange to say the least, how can it be that the virus was not inoculated, but the cells are heavily infected? I don't understand. If you mean that to facilitate the penetration of the virus you add trypsin - describe it, but there should still be a virus. Line 128, provides some clarity. Please use virology-appropriate terms.
- I have the same difficulty with the answer to question 6. I don't know what you mean. Please clarify somehow. There can't be no virus and the term mock-infected can be used. This is probably a cellular or some other control.
- Answer 11-if you created the figure, indicate below it in the appropriate way. Please check.
Author Response
comment 1: Line 75-79 - I understand that one result is new and the other is confirmatory?
Response:Thank you for your comment. A recent study has shown that the nsp5 protein of SADS-CoV inhibits IFN-β production, which was validated in this study. Additionally, this study reveals a novel finding: the nsp5 protein directly targets IKKε to reduce its protein abundance and phosphorylation level, thereby inhibiting IFN-β production. The molecular mechanism of IFN-β inhibition identified here represents an innovative discovery of this study.
comment 2: Regarding the answer to question 4: "The “mock-infected” means cells were treated with trypsin along without SADS-CoV inoculation"-This answer is strange to say the least, how can it be that the virus was not inoculated, but the cells are heavily infected? I don't understand. If you mean that to facilitate the penetration of the virus you add trypsin - describe it, but there should still be a virus. Line 128, provides some clarity. Please use virology-appropriate terms.
Response:Thank you for your comment. First, in the initial response, the term "mock-infection" has been amended to "control group". Given that trypsin is essential for viral replication, exogenous trypsin is consistently present in cell infection assays. To maintain strict single-variable control across experimental groups, the control group was defined as cells treated with trypsin alone (without viral inoculation). The experimental design is therefore structured as follows: Experimental group: Cells + trypsin + virus; Control group: Cells + trypsin. Notably, following incubation, experimental group cells exhibited pronounced cytopathic effects (CPE), and immunofluorescence staining with viral protein-specific antibodies revealed intense fluorescent signals. In contrast, control cells displayed no detectable CPE or fluorescent signals under identical culture duration and detection parameters. There is no such issue as "how can it be that the virus was not inoculated, but the cells are heavily infected?" as you pointed out. Furthermore, the manuscript has been revised to include: A detailed description of CPE and fluorescent signal analysis in the control group within Result 3.1; Elaboration on control group design in the Methods section (line 128).
comment 3: I have the same difficulty with the answer to question 6. I don't know what you mean. Please clarify somehow. There can't be no virus and the term mock-infected can be used. This is probably a cellular or some other control.
Response:Thank you for your comment. In the previous submission, "mock-infection" denoted cells treated with trypsin but without viral inoculation. There is no correlation between mock-infection and transfection—they represent different treatments in an experiment. For example, in Figure 2, to investigate whether SADS-CoV inhibits IFN-β production, we measured IFN-β expression (via promoter activity, mRNA levels, and protein levels) in both virus-infected and non-infected (mock-infection) cells. Since IFN-β expression was generally low under these conditions, researchers commonly stimulate cells with poly(I:C) (an IFN signaling agonist) to induce robust IFN-β production, allowing detection of whether SADS-CoV infection suppresses poly(I:C)–stimulated IFN-β levels. Thus, each panel in Figure 2 shows results from four experimental conditions: Cells + trypsin; Cells + trypsin + virus; Cells + poly(I:C) + trypsin; Cells + poly(I:C) + trypsin + virus. Importantly, "mock" solely indicates the absence of viral inoculation, while "transfection" refers to poly(I:C) transfection. In other experiments, "transfection" also denotes transfection of poly(I:C) or constructed plasmids.
comment 4 : Answer 11-if you created the figure, indicate below it in the appropriate way. Please check.
Response:Thank you for your comment. We have revised the legend of Figure 7 to "Figure 7. Schematic diagram of SADS-CoV nsp5 inhibiting interferon production by targeting kinase IKKε in this study."
Reviewer 3 Report
Comments and Suggestions for Authors
Most of the points I pointed out have been corrected well. Thanks for your hard work.
Author Response
commonts: Most of the points I pointed out have been corrected well. Thanks for your hard work.
response: Thank you for recognizing our research work.